# Digestibility of insect meals for Pacific white shrimp (*Litopenaeus vannamei*) and their performance for growth, feed utilization and immune responses

**Jaehyeong Shin**[1], **Kyeong-Jun Lee**[1,2]*

1 Department of Marine Life Science, Jeju National University, Jeju-si, Jeju Self-Governing Province, South Korea, 2 Marine Science Institute, Jeju National University, Jeju-si, Jeju Self-Governing Province, South Korea

* kjlee@jejunu.ac.kr

## Abstract

This study was conducted to examine digestibility of insect meals for Pacific white shrimp (*Litopenaeus vannamei*) and their utilization as fish meal substitutes. The tested insect meals were mealworm, silkworm, black soldier fly, rice grasshopper, two-spotted cricket, dynastid beetle and white-spotted flower chafer. Apparent digestibility coefficients of the tested insect meals were 83–89% for protein, 91–98% for lipid, 84–90% for energy, 77–81% for dry matter, 28–36% for chitin, 76–96% for amino acids and 89–93% for fatty acids. The amino acid availability of insect meals was high in taurine (93–96%), arginine (91–95%) and lysine (90–95%). Availability of fatty acids were 89–93% for saturated fatty acids, 90–93% for monounsaturated fatty acids and 88–93% for polyunsaturated fatty acids. For a feeding trial, a control diet was formulated using 27% tuna byproduct meal as a fish meal source and seven other diets were prepared replacing 10% tuna byproduct meal in the control diet with each insect meal. Triplicate groups of shrimp (initial body weight: 0.17 g) were fed the diets for 65 days. The growth performance was significantly improved when the shrimp were fed black soldier fly or dynastid beetle included diet. Dietary supplementation of insect meals significantly improved non-specific immune responses and antioxidant enzyme activity in the shrimp. These results indicate that the tested insect meals have high potentials to be used as a protein source that could replace fish meal in diets for the shrimp.

## Introduction

Insect meals have recently become an attractive alternative protein source for the production of sustainable aquaculture feeds [1]. In addition to their high protein levels, insects also rich in lipids, minerals and vitamins that support growth of shrimp and fish [2]. Insect larvae can rapidly convert low-quality organic wastes into high-quality fertilizer or growth promoters in animal feeds [3] and several species of insects have been found to possess antifungal and antibacterial properties [4]. The protein content of insects ranges from 50% to 82% (dry matter

**Data Availability Statement:** All relevant data are within the paper.

**Funding:** This work was supported by the National Research Foundation of Korea (NRF) grants

(2019R1A6A1A03033553 and NRF-
2018RID1A3B07046053). The funders had no role
in study design, data collection and analysis,
decision to publish, or preparation of the
manuscript.

**Competing interests:** The authors have declared
that no competing interests exist.

basis) depending on the insect species and/or their processing method [5]. Compared to most protein sources, insects farmed under controlled conditions could be a more viable protein source than fish meal (FM) in aquaculture feeds [1]. Insects are rich in essential amino acids (AAs) making them highly desirable as an excellent protein source for aquaculture [1]. Many insects have been reported to contain considerable amounts of taurine and hydroxyproline, both of which are lacking in plant protein sources [6]. Therefore, insect meals can be a promising protein source for the production of aquaculture feed. Several insect species used in fish feeds have also been reported to improve the immune response, antioxidant activity and disease resistance of aquatic animals [1]. Nonetheless, very few studies have assessed the benefits of using insect meals as a protein source for shrimps.

There are over a million known insect species worldwide and insects represent the largest and most diverse group within the Arthropoda phylum [7]. However, only a few insect species have been used for commercial purposes. The dynastid beetle (DB) (*Allomyrina dichotoma*) is a species of rhinoceros beetle that spends the majority of its life buried underground [8]. This species is native to East Asia and is widely used as a traditional medicine to treat many diseases [9]. The rice grasshopper (RG) (*Oxya chinensis*) is an oligophagous pest, primarily feeds on graminaceous grasses and has long been used as a food source in Asia [10,11]. The black soldier fly (BSF) (*Hermetia illucens*) is a true fly (Diptera) of the family Stratiomyidae and its larvae can consume materials such as food wastes and agricultural byproducts [12]. The white-spotted flower chafer (WFC) (*Protaetia brevitarsis*) is an important Scarabaeidae insect that is distributed throughout China and neighboring countries [13]. This insect has also been used in traditional East Asian medicine due to its excellent antithrombotic activity [14]. The mealworm (MW) (*Tenebrio molitor*) is commonly found in agricultural products and is considered the most promising species for commercial production and industrial applications [15]. The two-spotted cricket (TSC) (*Gryllus bimaculatus*) is considered a sporadic pest and also has a long history of traditional use in oriental medicine [16]. The silkworm (SW) (*Bombyx mori*) has long been reared worldwide for the production of silk and is currently used for the commercial production of medical or industrial biomaterials through genetic engineering [17].

Estimating the digestibility of a particular feed ingredient is the first step toward determining whether the ingredients in question can be used safely in fish and shrimp feeds [18]. The Pacific white shrimp (*Litopenaeus vannamei*) is the most widely cultured shrimp species, reaching a global production of 4 million tons in 2019 [19]. However, to the best of our knowledge, very few studies have evaluated the digestibility of insect meals for *L. vannamei* culture. Therefore, this study was conducted to examine the potential use of the above-described insects as protein sources for *L. vannamei* feed by evaluating their digestibility after which we conducted a feeding trial.

## Materials and methods

### Ethics statement

The protocols of digestibility test and feeding trial were evaluated and approved by Institutional Animal Care and Use Committee of Jeju National University (permit number: 2019–0039). Dissection was performed under ice anesthesia and all efforts were made to minimize suffering of shrimp.

### Test ingredients

The seven insect meals tested in this study were DB (Universal Farm's Meal Co. Ltd., Sunchang, Korea), RG (S-worm, Cheonan, Korea), BSF (CIEF Co. Ltd., Gimge, Korea), WFC (Universal Farm's Meal Co. Ltd., Sunchang, Korea), MW (KEIL Co. Ltd., Seoul, Korea), TSC

(Byeoli Co. Ltd., Goyang, Korea) and SW (Jamsil Farming Association, Suncheon, Korea). The insect meals were dried again and ground at a 200 μm size. Nutrient composition including chitin of the insect meals and tuna byproduct meal (TM) as a FM source were provided in Table 1.

**Table 1. Nutrient compositions (%, dry matter) of the seven insect meals and tuna byproduct meal for the digestibility and feeding trial of Pacific white shrimp (*Litopenaeus vannamei*).**

| Nutrient contents | Ingredients | | | | | | | |
|---|---|---|---|---|---|---|---|---|
| | DB | RG | BSF | WFC | MW | TSC | SW | TM |
| **Proximate composition** | | | | | | | | |
| Crude protein | 51.5 | 73.8 | 41.7 | 53.1 | 47.5 | 60.8 | 42.7 | 65.0 |
| Crude lipid | 22.9 | 6.19 | 17.4 | 17.6 | 30.8 | 20.1 | 7.53 | 9.24 |
| Crude ash | 3.61 | 7.76 | 18.7 | 4.17 | 4.27 | 4.99 | 11.5 | 15.4 |
| Moisture | 6.46 | 1.95 | 4.39 | 4.46 | 7.07 | 9.46 | 4.15 | 7.00 |
| **Essential amino acids (%, protein)** | | | | | | | | |
| Methionine | 0.30 | 0.16 | 0.27 | 0.30 | 0.23 | 0.27 | 0.26 | 1.35 |
| Lysine | 6.24 | 5.84 | 5.93 | 5.93 | 5.83 | 5.75 | 5.75 | 5.89 |
| Arginine | 4.74 | 6.58 | 5.36 | 4.22 | 5.23 | 6.71 | 5.02 | 4.71 |
| Histidine | 6.39 | 6.09 | 8.38 | 7.06 | 7.84 | 5.91 | 4.01 | 3.15 |
| Isoleucine | 5.39 | 4.79 | 4.78 | 4.44 | 5.00 | 4.74 | 4.50 | 3.64 |
| Leucine | 7.57 | 8.43 | 7.23 | 6.15 | 7.92 | 8.19 | 6.42 | 5.79 |
| Phenylalanine | 4.13 | 3.42 | 3.96 | 4.63 | 4.25 | 3.76 | 4.35 | 3.09 |
| Threonine | 4.33 | 3.89 | 4.15 | 4.16 | 4.12 | 3.96 | 4.18 | 3.42 |
| Valine | 6.83 | 6.93 | 6.70 | 5.61 | 7.09 | 6.84 | 5.66 | 4.11 |
| EAA/NAA[1] | 0.85 | 0.85 | 0.88 | 0.74 | 0.99 | 0.85 | 0.67 | 0.54 |
| **Fatty acids (%, lipid)** | | | | | | | | |
| C12:0 | 0.00 | 0.00 | 23.2 | 0.00 | 0.41 | 0.00 | 0.00 | 0.10 |
| C16:0 | 38.1 | 8.80 | 19.0 | 13.1 | 16.7 | 18.3 | 23.5 | 40.3 |
| C18:0 | 1.93 | 15.4 | 4.82 | 1.30 | 0.00 | 5.26 | 11.3 | 10.9 |
| C18:1n-9 | 48.9 | 21.3 | 23.4 | 60.3 | 43.4 | 28.9 | 26.3 | 4.50 |
| C18:2n-6 | 2.70 | 16.2 | 16.3 | 7.82 | 31.7 | 36.1 | 6.81 | 0.30 |
| C18:3n-3 | 0.00 | 33.5 | 2.18 | 0.30 | 1.36 | 9.60 | 29.5 | 0.30 |
| C20:5n-3 | 0.00 | 0.00 | 1.32 | 0.10 | 0.00 | 0.00 | 0.00 | 12.2 |
| C22:6n-3 | 0.00 | 0.00 | 0.26 | 0.00 | 0.00 | 0.00 | 0.00 | 7.90 |
| ΣSFA[2] | 41.0 | 29.0 | 52.8 | 15.9 | 21.5 | 24.7 | 36.4 | 66.6 |
| ΣMUFA[3] | 56.2 | 21.3 | 26.0 | 75.8 | 45.6 | 29.6 | 27.3 | 9.60 |
| ΣPUFA[4] | 2.87 | 49.7 | 20.3 | 8.38 | 33.1 | 45.7 | 36.3 | 21.8 |
| ΣPUFA n-3[5] | 0.00 | 33.5 | 3.76 | 0.40 | 1.36 | 9.60 | 29.5 | 20.4 |
| ΣPUFA n-6[6] | 2.87 | 16.2 | 16.6 | 7.98 | 31.7 | 36.1 | 6.81 | 1.40 |
| *n-3/n-6* | 0.00 | 2.07 | 0.23 | 0.05 | 0.04 | 0.27 | 4.33 | 14.6 |
| Chitin | 9.83 | 10.6 | 5.11 | 6.29 | 3.24 | 6.53 | 4.93 | 0.00 |

Ingredients are abbreviated as: dynastid beetle (DB), rice grasshopper (RG), black soldier fly (BSF), white-spotted flower chafer (WFC), mealworm (MW), two-spotted cricket (TSC), silkworm (SW) and tuna byproduct meal (TM) as a fish meal.

[1]Essential amino acid/non-essential amino acids.

[2]Sum of saturated fatty acids.

[3]Sum of monounsaturated fatty acids.

[4]Sum of polyunsaturated fatty acids.

[5]Sum of n-3 polyunsaturated fatty acids.

[6]Sum of n-6 polyunsaturated fatty acids.

## Digestibility test and feeding trial

For the digestibility test of the insect meals, a reference diet was formulated using TM and soybean meal as the major protein sources (Table 2). Each test ingredient was mixed with the reference diet at a ratio of 3:7 (w:w). Chromium oxide ($Cr_2O_2$, DaeJung Chemicals & Metals Co. Ltd., Siheung, Korea) was used by 1% in the reference diet as an inner indicator. The dry ingredients were mixed with cod liver oil and distilled water in a dough mixer and then the dough was pelleted (SP-50, Gumgang Engineering, Daegu, Korea) in 1–2 mm sizes. The pelleted diets were dried in a feed drier (SI-2400, Shinil General Dryer Co. Ltd., Daegu, Korea) at 25°C for 8 h. The diets were packed in zipper bags and stored at –25°C until use. The proximate composition, AAs, fatty acids and chitin level of the reference and test diets were provided in Table 3.

Shrimp post larvae were purchased from a local shrimp hatchery (Tamra shrimp, Jeju, Korea) and reared until they reached proper sizes. Total 264 shrimp (body weight: 5.15 ± 0.5 g) were distributed into eight 240 L capacity acrylic tanks. The shrimp were fed the reference diet for 6 days to be acclimated to the diets and tanks before the fecal collection for the digestibility. The average water temperature and dissolved oxygen (DO) were 27.8 ± 1.25°C and 5.04 ± 0.36 mg/L, respectively. Shrimp in each tank were fed one of the test diets two times

**Table 2. Dietary formulation and proximate composition of the reference diet (g/kg, dry matter) for the digestibility test of Pacific white shrimp (*Litopenaeus vannamei*).**

| Ingredients | g/kg diet |
|---|---|
| **Tuna byproduct meal[1]** | 250.0 |
| **Soybean meal** | 200.0 |
| **Squid liver meal** | 50.0 |
| **Wheat flour[2]** | 317.0 |
| **Starch** | 70.0 |
| **Cod liver oil[3]** | 30.0 |
| **Mineral premix[4]** | 20.0 |
| **Vitamin premix[5]** | 10.0 |
| **Mono-calcium phosphate** | 30.0 |
| **Lecithin[6]** | 10.0 |
| **Cholesterol** | 3.0 |
| **Chromium oxide[7]** | 10.0 |
| **Proximate compositions** | |
| **Crude protein** | 319.0 |
| **Crude lipid** | 78.9 |
| **Crude ash** | 125.0 |

[1]Tuna byproduct meal contains 60% crude protein. Woogin Feed Industry Co. Ltd., Incheon, Korea.

[2]Deahan Flour Co. Ltd., Incheon, Korea.

[3]E-wha oil & fat Industry Corp., Busan, Korea.

[4]Mineral premix (1 kg) contains 80 g $MgSO_4 \cdot 7H_2O$, 370 g $NaH_2PO_4 \cdot 2H_2O$, 130 g KCl, 40 g Ferriccitrate, 20 g $ZnSO_4 \cdot 7H_2O$, 356.64 g Ca-lactate, 0.2 g CuCl, 0.15 g $AlCl_3 \cdot 6H_2O$, 0.01 g $Na_2Se_2O_3$, 2 g $MnSO_4 \cdot H_2O$ and 1 g $CoCl_2 \cdot 6H_2O$.

[5]Vitamin premix (1 kg) contains 121 g L-ascorbic acid, 19 g DL-α tocopheryl acetate, 2.7 g thiamin hydrochloride, 9.1 g riboflavin, 1.8 g pyridoxine hydrochloride, 36 g niacin, 12.7 g Ca-D-pantothenate, 182 g myo-inositol, 0.27 g D-biotin, 0.68 g folic acid, 18 g p-aminobenzoic acid, 1.8 g menadione, 0.73 g retinyl acetate, 0.003 g cholecalciferol, 0.003 g cyanocobalamin and 594 g starch.

[6]Lysoforte™ Dry, KEMIN Korea Co. Ltd., Seongnam, Korea.

[7]DaeJung Chemicals & Metals Co. Ltd., Siheung, Korea.

**Table 3. Nutrient compositions (%, dry matter) of the reference diet (Ref) and test diets for the digestibility test of Pacific white shrimp (*Litopenaeus vannamei*).**

| | | Test diets (70% reference diet + 30% test ingredient) | | | | | | |
|---|---|---|---|---|---|---|---|---|
| | **Ref** | **DB** | **RG** | **BSF** | **WFC** | **MW** | **TSC** | **SW** |
| **Proximate composition (%)** | | | | | | | | |
| Dry matter | 94.4 | 94.7 | 94.2 | 94.4 | 94.6 | 94.9 | 94.9 | 94.4 |
| Crude protein | 31.9 | 39.7 | 45.4 | 36.4 | 40.8 | 38.7 | 42.5 | 37.4 |
| Crude lipid | 7.89 | 13.9 | 9.74 | 11.0 | 12.3 | 12.6 | 14.0 | 9.47 |
| Crude ash | 12.5 | 10.0 | 11.0 | 14.0 | 10.1 | 10.7 | 10.3 | 12.4 |
| Gross energy (kJ/g) | 16.3 | 18.0 | 16.9 | 16.7 | 17.6 | 17.6 | 17.9 | 16.6 |
| **Amino acids (%)** | | | | | | | | |
| Methionine | 0.65 | 0.77 | 0.87 | 0.69 | 0.79 | 0.72 | 0.83 | 0.69 |
| Lysine | 1.39 | 1.94 | 2.17 | 1.83 | 1.76 | 1.75 | 1.80 | 1.53 |
| Arginine | 2.15 | 2.81 | 3.59 | 2.82 | 2.79 | 3.04 | 3.01 | 2.68 |
| Histidine | 0.79 | 1.00 | 1.08 | 1.01 | 1.08 | 0.93 | 0.99 | 0.91 |
| Isoleucine | 1.29 | 1.68 | 1.83 | 1.51 | 1.55 | 1.54 | 1.70 | 1.47 |
| Leucine | 2.36 | 2.87 | 3.49 | 2.71 | 2.87 | 2.85 | 3.27 | 2.70 |
| Phenylalanine | 1.47 | 1.79 | 1.87 | 1.63 | 1.83 | 1.63 | 1.77 | 1.69 |
| Threonine | 1.32 | 1.69 | 1.93 | 1.59 | 1.75 | 1.57 | 1.79 | 1.63 |
| Valine | 1.48 | 2.00 | 2.27 | 1.88 | 1.92 | 1.93 | 2.20 | 1.81 |
| **Fatty acids (%)** | | | | | | | | |
| C12:0 | 0.00 | 0.00 | 0.19 | 1.08 | 0.00 | 0.00 | 0.00 | 0.00 |
| C14:0 | 0.20 | 0.32 | 0.19 | 0.40 | 0.14 | 0.36 | 0.16 | 0.17 |
| C16:0 | 1.82 | 3.05 | 1.88 | 2.46 | 2.30 | 2.43 | 2.84 | 2.17 |
| C16:1 | 0.25 | 0.42 | 0.24 | 0.30 | 0.86 | 0.30 | 0.24 | 0.21 |
| C18:0 | 0.72 | 1.04 | 1.01 | 0.92 | 0.73 | 0.83 | 1.13 | 0.98 |
| C18:1n-9 | 2.27 | 4.53 | 2.84 | 3.22 | 5.34 | 4.82 | 4.29 | 3.04 |
| C18:2n-6 | 1.69 | 3.02 | 1.90 | 1.93 | 2.21 | 3.20 | 3.90 | 1.66 |
| C18:3n-3 | 0.30 | 0.50 | 1.01 | 0.35 | 0.42 | 0.27 | 1.01 | 0.86 |
| C20:5n-3 | 0.19 | 0.31 | 0.15 | 0.14 | 0.10 | 0.13 | 0.13 | 0.11 |
| C22:6n-3 | 0.44 | 0.70 | 0.34 | 0.20 | 0.19 | 0.28 | 0.29 | 0.27 |
| **Chitin (%)** | 0.00 | 1.31 | 1.04 | 0.85 | 1.85 | 2.00 | 0.69 | 2.18 |

Test diets are abbreviated as: dynastid beetle (DB), rice grasshopper (RG), black soldier fly (BSF), white-spotted flower chafer (WFC), mealworm (MW), two-spotted cricket (TSC) and silkworm (SW).

(0830 and 1500 h) a day at a ratio of 3–4% body mass. Photoperiod was controlled by fluorescent lights on 13 h light and 11 h dark cycle. Uneaten diet and fecal residues in each tank were completely siphoned after each feeding. Feces were collected in two times (1100 and 1800 h) a day using a Pasteur-pipette for 7 days and combined as one replicate. The fecal collection lasted for 21 days to make triplicate samples per each diet. Collected feces were freeze dried for analyses of nutrients and chromium oxide.

For a feeding trial to verify the possibility of FM replacement, a control diet was formulated to meet the nutrient requirements for *L. vannamei* and seven other diets were prepared by substituting 10% TM with each insect meal (Table 4). The diets were prepared as described in the above. Total 720 shrimp (0.17 ± 0.00 g) were distributed into 24 acrylic tanks (240 L) in triplicates per dietary treatment. Shrimp were fed the diets four times (0830, 1100, 1400 and 1700 h) a day with a feeding rate of 3–10% of the biomass. The detailed feeding rate during the trial was as follows: 8–10% (0.17–2 g size), 5–7% (3–6 g size) and 3–4% (>7 g size). Total mass of shrimp in each tank was measured every two weeks to adjust the feeding rate. Seventy

**Table 4. Formulation and proximate composition (%, dry matter) of the experimental diets for the feeding trial of Pacific white shrimp (*Litopenaeus vannamei*).**

| Ingredients | Experimental diets | | | | | | | |
|---|---|---|---|---|---|---|---|---|
| | TM | DB | RG | BSF | WFC | MW | TSC | SW |
| TM | 27.0 | 17.0 | 17.0 | 17.0 | 17.0 | 17.0 | 17.0 | 17.0 |
| DB | 0.00 | 10.0 | 0.00 | 0.00 | 0.00 | 0.00 | 0.00 | 0.00 |
| RG | 0.00 | 0.00 | 10.0 | 0.00 | 0.00 | 0.00 | 0.00 | 0.00 |
| BSF | 0.00 | 0.00 | 0.00 | 10.0 | 0.00 | 0.00 | 0.00 | 0.00 |
| WFC | 0.00 | 0.00 | 0.00 | 0.00 | 10.0 | 0.00 | 0.00 | 0.00 |
| MW | 0.00 | 0.00 | 0.00 | 0.00 | 0.00 | 10.0 | 0.00 | 0.00 |
| TSC | 0.00 | 0.00 | 0.00 | 0.00 | 0.00 | 0.00 | 10.0 | 0.00 |
| SW | 0.00 | 0.00 | 0.00 | 0.00 | 0.00 | 0.00 | 0.00 | 10.0 |
| Casein | 0.70 | 2.60 | 0.00 | 4.20 | 2.40 | 2.90 | 1.50 | 4.10 |
| Soybean meal | 25.0 | 25.0 | 25.0 | 25.0 | 25.0 | 25.0 | 25.0 | 25.0 |
| Squid liver meal | 5.00 | 5.00 | 5.00 | 5.00 | 5.00 | 5.00 | 5.00 | 5.00 |
| Starch | 8.00 | 8.00 | 8.00 | 8.00 | 8.00 | 8.00 | 8.00 | 8.00 |
| Wheat flour | 23.1 | 23.2 | 23.8 | 19.6 | 23.1 | 23.9 | 23.3 | 19.7 |
| Soybean oil | 2.00 | 1.00 | 2.00 | 1.50 | 1.15 | 0.50 | 1.50 | 2.00 |
| Cod liver oil | 2.00 | 1.00 | 2.00 | 1.50 | 1.15 | 0.50 | 1.50 | 2.00 |
| Mineral mixture[1] | 2.00 | 2.00 | 2.00 | 2.00 | 2.00 | 2.00 | 2.00 | 2.00 |
| Vitamin mixture[2] | 1.00 | 1.00 | 1.00 | 1.00 | 1.00 | 1.00 | 1.00 | 1.00 |
| Lecithin | 1.00 | 1.00 | 1.00 | 1.00 | 1.00 | 1.00 | 1.00 | 1.00 |
| Cholesterol | 0.20 | 0.20 | 0.20 | 0.20 | 0.20 | 0.20 | 0.20 | 0.20 |
| Mono-calcium phosphate | 3.00 | 3.00 | 3.00 | 3.00 | 3.00 | 3.00 | 3.00 | 3.00 |
| Proximate composition | | | | | | | | |
| Crude protein | 38.5 | 38.5 | 38.7 | 38.3 | 38.2 | 38.5 | 38.9 | 38.7 |
| Crude lipid | 9.62 | 9.03 | 9.33 | 9.30 | 9.42 | 9.80 | 9.71 | 9.34 |
| Crude ash | 11.4 | 9.79 | 10.0 | 11.3 | 9.16 | 10.0 | 9.75 | 10.7 |

Experimental diets are abbreviated as: tuna byproduct meal (TM) as a fish meal, dynastid beetle (DB), rice grasshopper (RG), black soldier fly (BSF), white-spotted flower chafer (WFC), mealworm (MW), two-spotted cricket (TSC) and silkworm (SW).

[1]Mineral premix (1 kg) contains 80 g $MgSO_4 \cdot 7H_2O$, 370 g $NaH_2PO_4 \cdot 2H_2O$, 130 g KCl, 40 g Ferriccitrate, 20 g $ZnSO_4 \cdot 7H_2O$, 356.64 g Ca-lactate, 0.2 g CuCl, 0.15 g $AlCl_3 \cdot 6H_2O$, 0.01 g $Na_2Se_2O_3$, 2 g $MnSO_4 \cdot H_2O$ and 1 g $CoCl_2 \cdot 6H_2O$.

[2]Vitamin premix (1 kg) contains 121 g L-ascorbic acid, 19 g DL-α tocopheryl acetate, 2.7 g thiamin hydrochloride, 9.1 g riboflavin, 1.8 g pyridoxine hydrochloride, 36 g niacin, 12.7 g Ca-D-pantothenate, 182 g myo-inositol, 0.27 g D-biotin, 0.68 g folic acid, 18 g p-aminobenzoic acid, 1.8 g menadione, 0.73 g retinyl acetate, 0.003 g cholecalciferol, 003 g cyanocobalamin and 594 g starch.

percent of the rearing water volume in each tank was exchanged every three days. The water quality parameters were measured daily using Pro20 DO instrument (YSI, Yellow springs, OH, USA) and Seven Compact (Mettler Toledo, Columbus, OH, USA). Ammonia concentration was measured using the colorimetric method by Strickland and Parsons [20]. Average values of water quality were as follow: salinity (31 ± 1.25 ppt), DO (5.04 ± 0.30 mg/L), water temperature (27.8 ± 1.02°C), pH (7.82 ± 0.23) and ammonia (0.041 ± 0.025 mg/L).

## Sampling and analyses

After 65 days of the feeding trial, all the shrimp in each tank were weighed individually to calculate final body weight, specific growth rate, feed conversion ratio (FCR), protein efficiency ratio (PER) and survival. Five shrimp were captured from each tank and anesthetized in ice water. Shrimp hemolymph (0.2–0.3 ml per shrimp) was sampled using sterile syringes containing hank's balanced salt solution (55037C, Sigma-Aldrich, St. Louis, USA). Serum was

separated by centrifugation (Smart R17, Hanil Science Industrial Co. Ltd., Gimpo, Korea) at 700 x $g$ for 15 min and was stored at –60˚C for further analyses. The proximate compositions of ingredients, diets, feces and shrimp whole-body were analyzed according to methods of AOAC [21]. Protein was analyzed by the Kjeldahl method (Kjeltec™ 2300, FOSS analytical, Hilleroed, Denmark). Lipid was analyzed by Soxhlet extraction (SOX406 fat analyzer, Jinan Hanon Instruments, Shandong, China). Chromium oxide concentration in the feces and diets were determined by the method described by Divakaran et al. [22]. Apparent digestibility coefficients (ADCs) for the test and reference diets were calculated using the indicator method [23]:

$$ADCs\ (\%) = 100 - \left( \frac{\dfrac{\%\ indicator}{in\ diet}}{\dfrac{\%\ indicator}{in\ faeces}} \times \frac{\dfrac{\%\ nutrient}{in\ faeces}}{\dfrac{\%\ nutrient}{in\ diet}} \times 100 \right) \qquad (1)$$

where indicator is $Cr_2O_3$ and nutrient is dry matter, protein, lipid, energy, AAs, fatty acids and chitin. ADCs of nutrients in the test ingredients were calculated according to Cho et al. [24]:

$$ADCs\ (\%) = \left( \frac{100}{30} \right) \times \left[ ADC\ of\ test\ diet - \left( \frac{70}{100} \times ADC\ of\ reference\ diet \right) \right] \qquad (2)$$

The concentrations of AAs in the test ingredients, diets and feces were determined according to Ninhydrin method [25] using an AA analyzer (S433, Sykam GmbH, Fuerstenfeldbruck, Germany). Fatty acids were determined by a gas chromatography (68000GC, Agilent, Santa Clara, USA) based on Garces and Mancha [26]. Chitin was extracted and quantified according to Hackman [27] with a slight modification [28].

The activities of superoxide dismutase (SOD) and glutathione peroxidase (GPx) in shrimp hemolymph were measured with a commercial SOD assay kit (19160, Sigma-Aldrich, St. Louis, USA) and GPx assay kit (K762-100, Biovision, San Francisco, USA). The activity of phenoloxidase (PO) in hemolymph was measured by the method of Hernández-López et al. [29]. Nitro-blue tetrazolium (NBT) activity was analyzed based on Dantzler et al. [30].

### Statistical analysis

Data were analyzed with one-way analysis of variance (ANOVA) using SPSS version 17.0 (SPSS, Chicago, IL, USA). Duncan's multiple range test was used to find statistical differences among the experimental groups. Statistical significant differences were determined at $P < 0.05$.

### Results

In the digestibility test, protein ADC of the insect meals ranged from 83 to 89% (Table 5). DB showed relatively high ADC of protein. The protein ADC of SW was the lowest among all the tested insect meals. Lipid ADC was high in DB (98%) and MW (97.5%). The lowest lipid ADC was observed in SW. Energy ADC was ranged from 84 to 90% indicating relatively high values in DB and MW. Chitin ADC was ranged from 28 to 36%. ADC of AAs followed a similar pattern to protein ADC (Table 6). DB showed the highest ADC of AAs except for methionine. Methionine ADC was the highest in RG among all the insect meals. SW showed relatively low ADC of methionine, leucine, lysine, phenylalanine and threonine. ADC of AAs was high in taurine (93–96%) followed by arginine (91–95%) and lysine (90–95%). ADCs of fatty acids

**Table 5. Apparent digestibility coefficients (ADCs, %) of protein, lipid, energy, dry matter and chitin in the reference and insect meals for Pacific white shrimp (*Litopenaeus vannamei*).**

| Ingredients | Protein ADC | Lipid ADC | Energy ADC | Diet digestibility | Chitin digestibility |
|---|---|---|---|---|---|
| **Reference diet** | 89.2±1.62 | 91.7±2.29 | 88.2±2.61 | 78.1±4.20 | - |
| **DB** | 89.0±1.79[a] | 98.0±2.18[a] | 90.3±1.84 | 81.0±2.35[a] | 30.3±4.15 |
| **RG** | 86.3±2.80[ab] | 94.5±2.28[ab] | 87.5±4.08 | 80.7±1.77[a] | 33.1±7.39 |
| **BSF** | 85.1±5.58[ab] | 95.2±2.43[ab] | 87.1±6.08 | 78.5±3.52[ab] | 35.5±6.44 |
| **WFC** | 84.4±1.45[ab] | 94.0±2.00[ab] | 85.4±1.92 | 77.4±0.98[ab] | 28.3±5.66 |
| **MW** | 84.2±2.56[ab] | 97.5±1.02[ab] | 90.1±2.80 | 80.8±1.55[a] | 28.0±3.55 |
| **TSC** | 83.7±0.58[ab] | 95.0±3.62[ab] | 86.6±1.86 | 79.1±1.04[ab] | 30.3±4.15 |
| **SW** | 82.8±2.35[b] | 91.2±2.30[b] | 83.6±3.08 | 76.6±2.25[b] | 35.3±4.74 |

Values are mean of triplicates (n = 3) and presented as mean ± standard deviation. Different superscripts in each column indicate significant differences ($P < 0.05$). Ingredients are abbreviated as: dynastid beetle (DB), rice grasshopper (RG), black soldier fly (BSF), white-spotted flower chafer (WFC), mealworm (MW), two-spotted cricket (TSC) and silkworm (SW).

were 89–93% for saturated fatty acids (SFA), 90–93% for monounsaturated fatty acids (MUFA) and 88–93% for polyunsaturated fatty acids (PUFA) (Table 7).

After the feeding trial, growth was significantly higher in shrimp fed BSF and DB than that of shrimp fed the control diet (FM) (Table 8). FCR, PER and survival were not significantly affected by the diets. Proximate composition of whole-body did not show any significant difference among all the dietary groups (Table 9). Concentration of oleic acid (C18:1n-9) in

**Table 6. Apparent digestibility coefficients (ADCs, %) of essential and non-essential amino acids in the tested insect meals for Pacific white shrimp (*Litopenaeus vannamei*).**

| Amino acids | DB | RG | BSF | WFC | MW | TSC | SW |
|---|---|---|---|---|---|---|---|
| **Essential amino acids** | | | | | | | |
| **Methionine** | 91.9±2.15 | 93.0±1.51 | 88.6±1.01 | 90.1±1.22 | 90.6±0.84 | 91.5±1.12 | 87.0±0.48 |
| **Arginine** | 94.6±1.02 | 94.3±1.49 | 94.2±2.18 | 93.2±1.69 | 90.6±1.52 | 93.4±1.54 | 91.8±2.18 |
| **Histidine** | 91.7±1.56 | 89.0±0.86 | 91.1±1.69 | 88.8±1.02 | 90.3±2.09 | 87.9±2.50 | 89.1±1.25 |
| **Isoleucine** | 92.0±2.10 | 89.2±1.60 | 89.3±0.88 | 88.3±2.16 | 88.4±1.20 | 89.0±0.36 | 87.3±1.05 |
| **Leucine** | 92.1±0.61 | 89.5±2.04 | 89.9±1.44 | 89.5±1.29 | 88.4±2.49 | 89.3±1.16 | 88.4±2.82 |
| **Lysine** | 94.9±1.25 | 94.4±1.29 | 92.2±2.98 | 92.2±0.45 | 92.2±0.72 | 94.2±2.44 | 90.2±1.46 |
| **Phenylalanine** | 91.7±1.89 | 91.3±1.36 | 88.6±1.40 | 89.5±1.36 | 88.2±1.50 | 89.8±1.02 | 86.7±0.52 |
| **Threonine** | 90.0±2.06 | 89.6±0.76 | 87.1±2.23 | 87.6±0.98 | 84.8±2.43 | 87.4±2.17 | 84.8±1.69 |
| **Valine** | 89.0±0.84 | 83.0±1.67 | 87.2±1.99 | 84.1±2.42 | 85.2±1.45 | 83.6±0.63 | 84.4±2.41 |
| **Non-essential amino acids** | | | | | | | |
| **Taurine** | 95.0±0.15 | 96.4±1.42 | 94.1±1.58 | 94.8±1.11 | 94.1±0.29 | 96.2±1.01 | 92.8±1.25 |
| **Alanine** | 84.6±1.08 | 75.7±0.87 | 86.3±2.16 | 80.7±2.08 | 85.5±1.22 | 78.8±2.84 | 81.6±2.59 |
| **Aspartic acid** | 91.3±2.10 | 91.1±1.63 | 89.2±1.89 | 89.0±1.29 | 88.1±2.01 | 89.7±1.59 | 86.7±1.06 |
| **Glycine** | 87.7±1.01 | 84.5±0.49 | 84.2±0.78 | 85.6±0.71 | 85.3±1.68 | 84.1±0.65 | 81.1±0.42 |
| **Glutamic acid** | 93.5±0.59 | 93.5±2.46 | 91.4±1.46 | 92.2±1.52 | 89.6±2.89 | 92.5±1.49 | 89.8±1.14 |
| **Proline** | 94.4±1.42 | 85.8±1.65 | 91.2±2.03 | 90.6±0.88 | 88.5±1.49 | 87.9±2.57 | 89.2±2.18 |
| **Serine** | 90.9±1.39 | 89.2±2.07 | 87.5±1.52 | 89.1±1.23 | 83.7±2.12 | 86.9±1.23 | 84.5±1.69 |
| **Tyrosine** | 91.5±2.07 | 80.9±1.36 | 87.4±1.63 | 89.9±2.41 | 85.4±1.04 | 80.8±3.06 | 83.5±0.80 |

Values are mean of duplicates (n = 2) and presented as mean ± standard deviation.
Tested insect meals are abbreviated as: dynastid beetle (DB), rice grasshopper (RG), black soldier fly (BSF), white-spotted flower chafer (WFC), mealworm (MW), two-spotted cricket (TSC) and silkworm (SW).

**Table 7. Apparent digestibility coefficients (ADCs, %) of fatty acids in the tested insect meals for Pacific white shrimp (*Litopenaeus vannamei*).**

| Fatty acids | DB | RG | BSF | WFC | MW | TSC | SW |
|---|---|---|---|---|---|---|---|
| **Saturated fatty acids** | | | | | | | |
| C14:0 | 89.5±1.25 | 92.2±2.35 | 91.7±2.58 | 89.9±2.40 | 91.2±1.44 | 88.9±1.70 | 93.2±1.19 |
| C16:0 | 91.1±1.49 | 91.3±1.63 | 90.5±3.02 | 90.4±1.82 | 89.9±2.12 | 91.0±3.14 | 91.6±3.21 |
| C18:0 | 90.3±3.59 | 92.8±2.04 | 91.2±1.21 | 91.0±3.28 | 90.4±2.60 | 92.0±1.12 | 92.6±0.45 |
| **Monounsaturated fatty acids** | | | | | | | |
| C16:1 | 93.1±2.11 | 91.0±1.22 | 90.2±2.63 | 92.2±0.65 | 89.5±1.80 | 89.8±0.92 | 91.6±2.03 |
| C18:1n-9 | 91.7±1.30 | 90.8±3.16 | 90.7±1.47 | 92.0±1.55 | 91.7±3.15 | 91.5±1.86 | 91.8±3.20 |
| **Polyunsaturated fatty acids** | | | | | | | |
| C18:2n-6 | 89.8±0.71 | 90.8±1.77 | 89.5±2.42 | 89.7±2.22 | 90.9±1.41 | 91.5±2.77 | 91.2±1.85 |
| C18:3n-3 | 90.0±2.20 | 92.0±1.06 | 91.1±1.22 | 90.3±1.36 | 88.1±2.09 | 92.7±3.22 | 92.1±2.77 |

Values are mean of duplicates (n = 2) and presented as mean ± standard deviation.

Tested insect meals are abbreviated as: dynastid beetle (DB), rice grasshopper (RG), black soldier fly (BSF), white-spotted flower chafer (WFC), mealworm (MW), two-spotted cricket (TSC) and silkworm (SW).

shrimp muscle was numerically higher in DB, WFC, MW and TSC groups compared to the control group (Table 10). Eicosapentaenoic acid (EPA) and docosahexaenoic acid (DHA) levels in the muscle were lower in shrimp fed all the insect meal diets than those of shrimp fed the control diet. PO activity was significantly higher in shrimp fed BSF, RG and TSC than that of shrimp fed the control diet (Table 11). GPx activity was significantly higher in shrimp fed MW, BSF and TSC compared to that of the control group. NBT activity was significantly higher in shrimp fed RG, TSC, DB and WFC than that of shrimp fed the control diet. SOD activity did not differ among all the dietary groups.

## Discussion

This study was the first to determine the digestibility of various insect meals for the production of shrimp feeds. The ADC of the tested insect meals was 83–89% for protein, 91–98% for lipid,

**Table 8. Growth performance, feed utilization and survival of Pacific white shrimp (*Litopenaeus vannamei*) (initial body weight: 0.17 ± 0.00 g) fed the experimental diets for 65 days.**

| Dietary treatments | FBW[1] | SGR[2] | FCR[3] | PER[4] | FI[5] | Survival (%) |
|---|---|---|---|---|---|---|
| TM | 8.56±0.91[b] | 6.04±0.17[b] | 1.56±0.22 | 1.75±0.25 | 13.0±0.91[c] | 93.3±11.5 |
| DB | 11.1±1.26[a] | 6.41±0.16[a] | 1.36±0.17 | 1.92±0.23 | 14.8±0.88[ab] | 87.8±6.94 |
| RG | 9.71±0.08[ab] | 6.23±0.06[ab] | 1.50±0.06 | 1.72±0.07 | 14.3±0.71[ab] | 98.3±2.36 |
| BSF | 11.1±0.55[a] | 6.45±0.09[a] | 1.43±0.08 | 1.83±0.10 | 15.6±0.80[a] | 96.7±5.77 |
| WFC | 9.79±0.21[ab] | 6.23±0.02[ab] | 1.56±0.06 | 1.68±0.06 | 15.0±0.33[ab] | 93.3±3.33 |
| MW | 10.3±1.77[ab] | 6.31±0.27[ab] | 1.41±0.19 | 1.87±0.27 | 14.1±0.91[bc] | 92.2±10.7 |
| TSC | 10.3±0.82[ab] | 6.32±0.12[ab] | 1.45±0.12 | 1.78±0.14 | 14.6±0.37[ab] | 87.8±3.85 |
| SW | 9.69±0.57[ab] | 6.22±0.08[ab] | 1.58±0.08 | 1.64±0.08 | 15.0±0.13[ab] | 94.4±5.09 |

Values are mean of triplicates (n = 3) and presented as mean ± standard deviation. Different superscripts in each column indicate significant differences ($P < 0.05$).

Dietary treatments are abbreviated as: tuna byproduct meal (TM) as a fish meal, dynastid beetle (DB), rice grasshopper (RG), black soldier fly (BSF), white-spotted flower chafer (WFC), mealworm (MW), two-spotted cricket (TSC) and silkworm (SW).

[1] Final body weight (g).

[2] Specific growth rate (%) = [(log$_e$ final body weight – log$_e$ body weight) ÷ days] × 100.

[3] Feed conversion ratio = feed intake ÷ wet weight gain.

[4] Protein effiency ratio = wet weight gain ÷ total protein given.

[5] Feed intake (g) = dry feed consumed (g) ÷ the number of fish.

**Table 9. Whole-body composition (%, wet basis) of Pacific white shrimp (*Litopenaeus vannamei*) fed the experimental diets for 65 days.**

| Dietary treatments | Crude protein | Crude lipid | Crude ash | Moisture |
|---|---|---|---|---|
| TM | 18.9±0.31 | 1.34±0.40 | 3.78±0.13 | 76.2±0.64 |
| DB | 19.3±0.12 | 1.33±0.18 | 3.56±0.25 | 77.0±0.42 |
| RG | 19.7±0.30 | 1.27±0.11 | 3.53±0.15 | 76.2±0.25 |
| BSF | 19.5±0.15 | 1.35±0.13 | 3.40±0.23 | 76.5±0.37 |
| WFC | 19.4±0.35 | 1.37±0.10 | 3.59±0.28 | 76.7±0.65 |
| MW | 19.5±0.83 | 1.44±0.14 | 3.35±0.28 | 76.3±0.40 |
| TSC | 19.0±0.19 | 1.37±0.15 | 3.63±0.15 | 76.2±0.21 |
| SW | 19.4±0.20 | 1.40±0.12 | 3.53±0.16 | 76.5±0.27 |

Values are mean of triplicates (n = 3) and presented as mean ± standard deviation. Different superscripts in each column indicate significant differences ($P < 0.05$). Dietary treatments are abbreviated as: tuna byproduct meal (TM) as a fish meal, dynastid beetle (DB), rice grasshopper (RG), black soldier fly (BSF), white-spotted flower chafer (WFC), mealworm (MW), two-spotted cricket (TSC) and silkworm (SW).

84–90% for energy, 77–81% for dry matter, 76–96% for AAs and 89–98% for fatty acids. Panini et al. [31] reported that the ADC of MW in *L. vannamei* was 76.1% for protein, 66.5% for energy, 45.9% for dry matter and 72–86% for AAs showing relatively low values compared to our results. In the present study, DB had the highest protein, lipid and energy ADC in *L. vannamei* (84–92%, 92% and 87–97%, respectively). Furthermore, lipid ADC of the tested insect meals was relatively higher than those obtained from FM in previous studies [32–34]. The availability of lysine and methionine, the two most limiting AAs in the plant protein sources,

**Table 10. Fatty acid composition (%, lipid) of muscle of Pacific white shrimp (*Litopenaeus vannamei*) fed the experimental diets for 65 days.**

| Fatty acids | TM | DB | RG | BSF | WFC | MW | TSC | SW |
|---|---|---|---|---|---|---|---|---|
| **Saturated fatty acids (SFA)** | | | | | | | | |
| C16:0 | 25.0±2.12 | 29.8±1.44 | 25.2±1.23 | 25.4±2.88 | 25.2±1.58 | 25.8±1.52 | 25.9±2.10 | 25.5±2.22 |
| C18:0 | 19.6±3.01 | 15.8±2.87 | 19.8±2.27 | 20.5±1.23 | 16.6±2.36 | 17.7±2.55 | 18.0±1.26 | 21.1±1.10 |
| **Monounsaturated fatty acids (MUFA)** | | | | | | | | |
| C18:1n-9 | 17.5±2.44 | 26.8±2.36 | 18.0±3.10 | 18.5±2.24 | 22.9±3.00 | 23.0±2.10 | 19.2±2.03 | 18.5±2.14 |
| **Polyunsaturated fatty acids (PUFA)** | | | | | | | | |
| C18:2n-6 | 17.6±1.17 | 13.7±2.02 | 18.8±2.33 | 17.5±3.25 | 18.0±2.19 | 18.2±1.01 | 20.5±1.28 | 17.7±2.42 |
| C20:5n-3 | 10.8±2.39 | 7.89±1.42 | 10.4±1.14 | 10.4±0.65 | 9.18±1.41 | 8.19±0.84 | 9.02±1.20 | 9.96±1.03 |
| C22:6n-3 | 9.47±1.21 | 5.96±1.66 | 7.70±0.67 | 7.73±1.56 | 8.03±1.88 | 7.21±1.11 | 7.38±0.68 | 7.27±1.23 |
| ΣSFA[1] | 44.6±3.10 | 45.6±2.18 | 45.0±3.58 | 45.9±2.46 | 41.9±2.36 | 43.5±3.25 | 43.9±2.86 | 46.6±3.02 |
| ΣMUFA[2] | 17.5±2.44 | 26.8±2.36 | 18.0±3.10 | 18.5±2.24 | 22.9±3.00 | 23.0±2.10 | 19.2±2.03 | 18.5±2.14 |
| ΣPUFA[3] | 37.9±3.12 | 27.6±2.55 | 37.0±1.15 | 35.6±2.12 | 35.2±3.02 | 33.6±2.48 | 36.9±1.63 | 34.9±2.06 |
| ΣPUFA n-3[4] | 20.3±2.56 | 13.8±1.56 | 18.1±2.48 | 18.1±1.85 | 17.2±2.69 | 15.4±1.21 | 16.4±2.16 | 17.2±1.85 |
| ΣPUFA n-6[5] | 17.6±1.17 | 13.7±2.02 | 18.8±2.33 | 17.5±3.25 | 18.0±2.19 | 18.2±1.01 | 20.5±1.28 | 17.7±2.42 |
| *n-3/n-6* | 1.15±0.16 | 1.01±0.18 | 0.96±0.08 | 1.04±0.18 | 0.96±0.16 | 0.85±0.12 | 0.80±0.16 | 0.98±0.17 |

Values are mean of duplicates (n = 2) and presented as mean ± standard deviation.

Dietary treatments are abbreviated as: tuna byproduct meal (TM) as a fish meal, dynastid beetle (DB), rice grasshopper (RG), black soldier fly (BSF), white-spotted flower chafer (WFC), mealworm (MW), two-spotted cricket (TSC) and silkworm (SW).

[1]Sum of saturated fatty acids.

[2]Sum of monounsaturated fatty acids.

[3]Sum of polyunsaturated fatty acids.

[4]Sum of n-3 polyunsaturated fatty acids.

[5]Sum of n-6 polyunsaturated fatty acids.

**Table 11. Innate immune responses and antioxidant enzyme activities of Pacific white shrimp (*Litopenaeus vannamei*) fed the experimental diets for 65 days.**

| Dietary treatments | PO[1] (absorbance) | SOD[2] (% inhibition) | GPx[3] (mU/ml) | NBT[4] (absorbance) |
|---|---|---|---|---|
| TM | 0.17±0.01$^c$ | 99.8±0.47 | 80.2±4.91$^b$ | 0.55±0.04$^b$ |
| DB | 0.18±0.01$^{bc}$ | 96.4±4.25 | 106±9.21$^a$ | 0.64±0.06$^{ab}$ |
| RG | 0.19±0.02$^{abc}$ | 99.3±1.73 | 96.0±6.05$^{ab}$ | 0.61±0.08$^b$ |
| BSF | 0.21±0.03$^{ab}$ | 100±0.20 | 105±3.34$^a$ | 0.64±0.08$^{ab}$ |
| WFC | 0.21±0.02$^{ab}$ | 99.0±1.82 | 85.3±1.42$^{ab}$ | 0.74±0.02$^a$ |
| MW | 0.22±0.03$^a$ | 98.7±1.63 | 104±6.75$^a$ | 0.72±0.14$^a$ |
| TSC | 0.19±0.01$^{bc}$ | 99.9±1.29 | 82.3±1.71$^b$ | 0.72±0.10$^a$ |
| SW | 0.21±0.03$^{abc}$ | 99.0±0.43 | 99.6±4.77$^{ab}$ | 0.74±0.12$^a$ |

Values are mean of triplicates (n = 3) and presented as mean ± standard deviation. Different superscripts in each column indicate significant differences ($P < 0.05$). Dietary treatments are abbreviated as: tuna byproduct meal (TM) as a fish meal, dynastid beetle (DB), rice grasshopper (RG), black soldier fly (BSF), white-spotted flower chafer (WFC), mealworm (MW), two-spotted cricket (TSC) and silkworm (SW).

[1]Phenoloxidase

[2]Super oxide dismutase

[3]Glutathione peroxidase

[4]Nitro-blue tetrazolium.

of the tested insect meals, were 90–95% and 87–93%, respectively, which was consistent with previous studies where lysine and methionine availability of FM were ranged 92.0–92.7% and 93.9–94.7%, respectively [32,34]. Interestingly, all the tested insect meals had very high taurine availability, which could be another advantage of using the insect meals in shrimp feeds. The insect meals evaluated in this study exhibited better digestibility than other previously reported protein sources. Our findings indicated that insect meals can be used as highly digestible protein sources in shrimp feed.

In this study, chitin digestibility of the tested insect meals was relatively lower (28–36%) compared to other nutrients. Chitin is an unbranched polysaccharide and a major constituent of insect exoskeletons, which is composed of N-acetylglucosamine and glucosamine [1]. Clark et al. [35] reported that the digestibility of dietary crustacean chitin was very low in adult-sized *L. vannamei* (17 g; 36%), Atlantic white shrimp (*L. setiferus*) (35 g; 33%) and pink shrimp (*Farfantepenaeus duorarum*) (17 g; 52%). The low chitin digestibility of the tested insect meals in the present study might have resulted from a limited ability of the shrimp to synthesize chitinase *in vivo*. Rocha et al. [36] detected two chitinase isoenzymes in *L. vannamei* hepatopancreas. Further, Tzuc et al. [37] identified several chitinase-secreting microorganisms in the digestive tract of *L. vannamei*, which enabled the partial digestion of the dietary chitin. Therefore, dietary insect chitin could partly be digested by *L. vannamei*. Chitin is another bioactive compound in insects. Dietary supplementation with crustacean chitin improved the growth of black tiger shrimp (*Penaeus monodon*) [38,39] and giant freshwater prawn (*Macrobrachium rosenbergii*) [40], in addition to enhancing the resistance of shore crab (*Carcinus maenas*) against *Vibrio alginolyticus* [41] and reducing oxidative stress in *P. monodon* [42]. Many studies have reported that the structure of insect chitin is similar to that of crustacean chitin and some insect chitin have relatively high chitinase affinity compared to crustacean chitin [43]. Henry et al. [44] hypothesized that insect chitin may also have immunostimulant properties when incorporated into feeds for European sea bass (*Dicentrarchus labrax*). Insect chitin is likely to contribute directly or indirectly to the immune responses of the shrimp. Nonetheless, dietary chitin levels should be carefully optimized, as excessive dietary chitin supplementation (>10%) reduces growth, feed utilization and digestibility (protein and lipid) in *P. monodon* [39].

In the feeding trial, dietary replacement of 17% FM with each insect meal did not show any significantly reduced growth or feed utilization of *L. vannamei*. On the contrary, the insect meal inclusions (or replacement of FM) in the diets improved the growth performance of the shrimp. Cummins et al. [45] examined the dietary utilization of BSF for *L. vannamei* (1.2–16 g) and found that BSF can replace 25% FM without any negative effect. Motte et al. [46] reported that *L. vannamei* fed a diet in which 50% FM was replaced with a defatted MW had significantly higher weight gain and lower FCR than a control group (no replacement). MW was also reported to replace up to 50% FM [47] without any significant impairment. Rahimnejad et al. [48] reported that high dietary SW levels enhanced dry matter and protein digestibility in *L. vannamei*. A review article by Henry et al. [49] indicated that insects have great potential as a protein source in fish feeds due to their good AA profiles and high levels of taurine and hydroxyproline compared to most plant protein sources. Another possibility for the improved growth performance in the present study might be due to a certain level of chitin, antimicrobial peptides (AMPs) and unknown growth factors in the insects which have positively contributed to growth performance, feed utilization and immune responses of fishes [1,44]. Recent studies also indicated that intestinal microbial communities, anti-inflammatory factors and digestive enzyme activity of fishes could be enhanced by the inclusion of insect proteins in their diets [50–52].

In the present study, the tested insect meals appeared to meet all the nutritional requirements of *L. vannamei*, at least at the tested FM replacement levels. More importantly, the proximate composition of the whole-body shrimp was not influenced by the inclusion of the tested insect meals, which are consistent with previous studies on SW [48] or BSF [45]. In contrast, Panini et al. [31] reported that feeding *L. vannamei* with non-defatted MW increased their whole-body lipid content. These discrepancies may be due to variations in the nutrient contents of the insect meals as well as processing methods and life cycle stage of the insects and their dietary formulations [53].

Our findings also revealed that dietary supplementation of the tested insect meals can improve the innate immune responses and antioxidant enzyme activities of *L. vannamei*. Insects are known to contain a variety of AMPs, which possesses several health-promoting properties including antibiotic activity [54]. Jozefiak and Engberg [4] emphasized that insects could be a promising source due to their AMPs which could be used as an alternative to antibiotics in animal feeds. For the AMPs, MW is known to contain tenascin 1 [55], BSF and WSFC contain defensin-like peptides [56,57], DB contains defensin [58] and SW contains moricin [59] and sericin [60]. Insect AMPs can disrupt bacteria membranes [4]. Certain AMPs can pass through the membrane and thereby interfere with DNA or RNA synthesis of the host bacteria [61]. Choi et al. [47] reported that dietary supplementation of MW enhanced the immune responses of *L. vannamei* by upregulating expression of β-1, 3-glucan binding protein, prophenoloxidase and crustin genes. Dietary MW supplementation also enhanced the resistance of *L. vannamei* to *V. parahaemolyticus*, the causative pathogen of early mortality syndrome [46]. Motte et al. [46] explained that one of the increased immunological benefits of MW (e.g., disease resistance) with its AMP contents. Therefore, the AMPs present in the tested insect meals could explain the enhanced innate immunity and antioxidant enzyme activity of the shrimp in the current study. A study is needed to confirm the effects of dietary supplementation of insect AMPs on shrimp immune responses and physiological activities.

Dietary inclusion of the tested insect meals also affected fatty acid composition of the shrimp muscle, which mirrored the fatty acid composition of the diets. The fatty acid profiles of the tested insect meals exhibited relatively high levels of MUFA (18:1n-9) and PUFA (18:2n-6 and 18:3n-3) and low levels or a lack of highly unsaturated fatty acids (DHA and EPA) compared with FM (Table 1). Many studies have reported that dietary inclusion of insect meals

increases MUFA concentration and decreases highly unsaturated fatty acids (DHA and EPA) levels in muscle of shrimp [31] and fish [62]. Terrestrial insects lack DHA and EPA, which is attributed to their low Δ-5 and Δ-6 desaturase activities [63]. However, the fatty acid composition of insects can be affected by the nutrient contents of their feeds during their growing stages [49]. Liland et al. [64] reported that n-3 fatty acid supplementation in diets for BSF larvae improved DHA and EPA levels in their fatty acid compositions. Therefore, the fatty acid profiles of insects are thought to be easily modified, which is another benefit of using insect proteins in aquaculture feeds. Lauric acid is the most abundant compound in fatty acid profile of BSF (Table 1). Belghit et al. [65] reported that dietary supplementation of BSF oil decreased liver triacylglycerol levels in Atlantic salmon (*Salmo salar*) suggesting that lauric acid content of BSF may act as a low-deposition and rapid-oxidation medium-chain fatty acid (MCFA) in the aquatic animals. In addition to their rapid absorption and oxidation, MCFAs have also been reported to possess antimicrobial and antiviral properties [66–68]. MCFA transportation is less dependent on chylomicrons and lipoproteins due to higher polarity of MCFAs compared to long-chain fatty acids. Therefore, the oil fraction and bioactive compounds in BSF could be a reason for the enhanced shrimp growth and immune responses observed in this study.

## Conclusions

Our findings demonstrate the applicability of the tested insect meals as protein sources for the production of *L. vannamei* feed. The dietary supplementation or FM replacement with the tested insect meals could improve the innate immunity and antioxidant capacity of the shrimp. Further studies are needed to characterize the properties of the bioactive compounds contained in the insect meals and to assess their effects and safety.

## Author Contributions

**Investigation:** Jaehyeong Shin.

**Supervision:** Kyeong-Jun Lee.

**Writing – original draft:** Jaehyeong Shin.

**Writing – review & editing:** Kyeong-Jun Lee.

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
