## [Decision Letter · Decision Letter 0]

17 Aug 2021

PONE-D-21-14702

Digestibility of insect meals for Pacific white shrimp (Litopenaeus vannamei) and their performance for growth, feed utilization and immune responses

PLOS ONE

Dear Dr. Lee,

Thank you for submitting your manuscript to PLOS ONE. After careful consideration, we feel that it has merit but does not fully meet PLOS ONE’s publication criteria as it currently stands. Therefore, we invite you to submit a revised version of the manuscript that addresses the points raised during the review process.

We look forward to receiving your revised manuscript.

Kind regards,

Mahmoud A.O. Dawood, PhD

Academic Editor

PLOS ONE

1. Please ensure that your manuscript meets PLOS ONE's style requirements, including those for file naming. The PLOS ONE style templates can be found at https://journals.plos.org/plosone/s/file?id=wjVg/PLOSOne_formatting_sample_main_body.pdf and https://journals.plos.org/plosone/s/file?id=ba62/PLOSOne_formatting_sample_title_authors_affiliations.pdf.

We will update your Data Availability statement on your behalf to reflect the information you provide."

Additional Editor Comments (if provided):

Dear authors,

The reviewers listed some revisions to your submission. Besides I suggest you the following revisions:

All tables should stand alone with full definition of all abbreviations.

Make sure that all works cited in the text are in the reference list, that the presentation is consistent and that correct information is given.

Define and explain all acronyms and abbreviations on first mention in the text.

On first mention of a species in the text give both the common (trivial) and formal name, and make sure that the presentation is correct and consistent.

Make sure that symbols, sub- and super-scripts, upper- and lower-case are presented correctly, and that there is correct and consistent use of italics, brackets and punctuation etc.

There are mistakes in the reference list, including incorrect reporting, inconsistent presentation, spelling mistakes and problems with use of punctuation etc.

Reviewers' comments:

Reviewer's Responses to Questions

**Comments to the Author**

1. Is the manuscript technically sound, and do the data support the conclusions?

Reviewer #1: Yes

Reviewer #2: Yes

2. Has the statistical analysis been performed appropriately and rigorously? 

Reviewer #1: Yes

Reviewer #2: Yes

3. Have the authors made all data underlying the findings in their manuscript fully available?

Reviewer #1: Yes

Reviewer #2: No

4. Is the manuscript presented in an intelligible fashion and written in standard English?

Reviewer #1: Yes

Reviewer #2: No

5. Review Comments to the Author

Reviewer #1: This research is timely and will add a lot to current knowledge on insect meals. I believe that this will be greatly accepted in the community of shrimp aquaculture. There are a few minor revisions suggested, as outlined below:

On line 19, wrong conjugation, change "was" to "were".

Line 24-25, it is not clearly understood how insect meals were included, please rephrase to reflect that 17% of tuna by-product meal was supplemented with 10% insect meal.

Line 38, "Insect also provides" or "insects also provide".

Line 42, "day matter", please change this.

Line 47, How do the insect EAA profiles significantly differ from fish meal? What do previous findings show on this matter?

Line 144- "Seven other diets"

Line 148, is there a particular reason for the four feedings and also the range in feeding rate (3-10%)? Are they fed more as the trial progresses or are they fed more based on observed feed intake?

Line 150, please remove the "in" before every three days.

Line 151, please change "measure" to "measured".

Line 316, "been" should be removed.

Line 330-331, what makes you say that excess dietary chitin can impair growth and feed utilization? Has there been a study to back this or is it speculation based on experience that excess of most things tend to be detrimental to the cultured species' health?

Line 332, I am unclear what is meant by this statement, it seems to mean that 37% of FM was completely replaced and not supplemented with insect meal, please re-word or please explain what is meant here.

Line 359, I do believe that there is more than one finding, so the correct term should be "findings", even if you are only referring to that particular section.

Reviewer #2: This study was conducted to examine digestibility of insect meals for Pacific white

16 shrimp (Litopenaeus vannamei) and their utilization as fish meal substitutes. This manuscript has innovation.

áManuscript must be check for grammar structure by native person. discussion part should be modified and use newer references. M@M must explain in more details.

6. PLOS authors have the option to publish the peer review history of their article (what does this mean?). If published, this will include your full peer review and any attached files.

Reviewer #1: **Yes: **Dr. Amina Moss

Reviewer #2: No

---

## [Author Response · Author response to Decision Letter 0]

7 Sep 2021

Dear Editor In-Chief,

Thank you very much for the reviewer’s comments to improve our manuscript (PONE-D-21-14702). According to the reviewer’s comments/suggestions including yours, all the required changes or revisions were carefully made in the revised manuscript. The main corrections and answers are listed below:

[Editor In-Chief]

Q1: All tables should stand alone with full definition of all abbreviations.

A1: We revised all the Tables as the editor suggested. Please find the revised Tables in the revised manuscript.

Q2: Make sure that all works cited in the text are in the reference list, that the presentation is consistent and that correct information is given.

A2: We carefully checked all the references and corrected accordingly. 

Q3: Define and explain all acronyms and abbreviations on first mention in the text.

A3: We carefully revised the manuscript and confirmed that all the acronyms and abbreviations are fully explained in its first mention.

Q4: On first mention of a species in the text give both the common (trivial) and formal name, and make sure that the presentation is correct and consistent.

A4: We carefully revised the manuscript and corrected the mistakes.

Q5: Make sure that symbols, sub- and super-scripts, upper- and lower-case are presented correctly, and that there is correct and consistent use of italics, brackets and punctuation etc.

A5: We corrected and made them constant. Please check the revised manuscript. 

Q6: There are mistakes in the reference list, including incorrect reporting, inconsistent presentation, spelling mistakes and problems with use of punctuation etc.

A6: We carefully revised and corrected the reference list including mistakes along with journal’s requirements. 

[Reviewer #1]

Q1: Line 19. Wrong conjugation, change "was" to "were".

A1: We corrected the sentence as the reviewer suggested (Line 19).

Q2: Line 24-25. It is not clearly understood how insect meals were included, please rephrase to reflect that 17% of tuna by-product meal was supplemented with 10% insect meal.

A2: We agree to the reviewer’s comment. We modified the sentence as follows; ‘For a feeding trial, a control diet was formulated using 27% tuna byproduct meal as a fish meal source and seven other diets were prepared replacing 10% tuna byproduct meal in the control diet with each insect meal (10%).’ (Line 24–26).

Q3. Line 38. "Insect also provides" or "insects also provide".

A3: We corrected it as the reviewer suggested (Line 36).

Q4. Line 42. "day matter", please change this.

A4: We revised the typo-error in the sentence as the reviewer suggested (Line 40).

Q5. Line 47. How do the insect EAA profiles significantly differ from fish meal? What do previous findings show on this matter?

A5: According to many previous studies, fish meal was the best ingredient for fishes and shrimps because of its excellent essential amino acid balance. The essential amino acid balance in the insects has also been reported to be comparable to those of fish meal with minor difference (Nogales‐Mérida et al., 2019). Therefore, we changed the sentence to ‘Insects are rich in essential amino acids (AAs) making them highly desirable as an excellent protein sources for aquaculture [1]’. (Line 43–44).

Q6. Line 144. "Seven other diets"

A6: We corrected the sentence as the reviewer suggested (Line 150).

Q7. Line 148. Is there a particular reason for the four feedings and also the range in feeding rate (3-10%)? Are they fed more as the trial progresses or are they fed more based on observed feed intake?

A7: The feeding times of four have been the most frequently used feeding time for shrimp (Tacon et al., 2002). The average feeding rate for shrimp trials has long been approximately 2–10% of total body mass (Tacon et al., 2002). We fed the shrimp in the trial with the amounts as follows; 8–10% (0.17–2 g), 5–7% (3–6 g) and 3–4% (>7 g) based on the growth rate of shrimp. The feeding rate was reduced as the trial progressed. We added the details in the revised manuscript (Line 155–156).

References: Tacon AGJ, Cody JJ, Conquest LD, Divakaran S, Forster IP, Decamp OE. Effect of culture system on the nutrition and growth performance of Pacific white shrimp (Litopenaeus vannamei Boone) fed different diets. Aquaculture nutrition 2002; 8:121–137.

Q8. Line 150. Please remove the "in" before every three days.

A8: We changed the sentence as the reviewer suggested (Line 157).

Q9. Line 151. Please change "measure" to "measured".

A9: We changed the sentence as the reviewer suggested (Line 158).

Q10. Line 316. "been" should be removed.

A10: We changed the sentence as the reviewer suggested (Line 321).

Q11. Line 330-331. What makes you say that excess dietary chitin can impair growth and feed utilization? Has there been a study to back this or is it speculation based on experience that excess of most things tend to be detrimental to the cultured species' health?

A11: It was reported that dietary chitin supplementation at proper levels (2–5% in diet) enhances growth, feed utilization and digestibility of shrimp (Shiau and Yu, 1998). However, an excessive dietary chitin supplementation (10% in diet) had a negative effect on their growth, feed utilization and digestibility (protein and lipid) in the study. We modified the sentence as follows: ‘Nonetheless, dietary chitin levels should be carefully optimized, as excessive dietary chitin supplementation (>10% in diet) reduced growth, feed utilization and digestibility (protein and lipid) of P. monodon [39].’ (Line 335–337).

References: [39] Shiau SY, Yu YP. Chitin but not chitosan supplementation enhances growth of grass shrimp, Penaeus monodon. J Nutr. 1998; 128:908–912.

Q12. Line 332. I am unclear what is meant by this statement, it seems to mean that 37% of FM was completely replaced and not supplemented with insect meal, please re-word or please explain what is meant here.

A12: We agree with the reviewer’s comment. The sentence could make the readers confused even though fish nutritionists usually use the actual replacement percentage from FM, which is 37% fish meal replacement in this study for 10% fish meal in the diet (10/27 =0.37) with 10% insect meal. Nonetheless, we modified the sentence to improve readability or understanding as the reviewer suggested as follow; ‘In the feeding trial, dietary replacement of 17% FM with each insect meal did not show any significantly reduced growth or feed utilization of L. vannamei.’ (Line 338–339).

Q13. Line 359. I do believe that there is more than one finding, so the correct term should be "findings", even if you are only referring to that particular section.

A13: Thank you for the suggestion. We corrected the sentence accordingly (Line 365).

[Reviewer #2]

Q1. Manuscript must be check for grammar structure by native person.

A1. The manuscript was rechecked by an English native (essayreview.co.kr). We carefully revised the original manuscript to improve readability. Please find the revised manuscript.

Q2. Discussion part should be modified and use newer references.

A2. We agree with the reviewer’s comment. We carefully revised some sentences accordingly in discussion part. Please check the revised manuscript.

Q3. Material and methods must explain in more details.

A3. We agree to the reviewer’s comment. We added more details in material and methods section including detailed feeding rate. Please check the revised manuscript.

---

## [Decision Letter · Decision Letter 1]

11 Oct 2021

PONE-D-21-14702R1Digestibility of insect meals for Pacific white shrimp (Litopenaeus vannamei) and their performance for growth, feed utilization and immune responsesPLOS ONE

Dear Dr. Jun Lee,

Thank you for submitting your manuscript to PLOS ONE. After careful consideration, we feel that it has merit but does not fully meet PLOS ONE’s publication criteria as it currently stands. Therefore, we invite you to submit a revised version of the manuscript that addresses the points raised during the review process.

We look forward to receiving your revised manuscript.

Kind regards,

Mahmoud A.O. Dawood, PhD

Academic Editor

PLOS ONE

Journal Requirements:

Reviewers' comments:

Reviewer's Responses to Questions

**Comments to the Author**

1. If the authors have adequately addressed your comments raised in a previous round of review and you feel that this manuscript is now acceptable for publication, you may indicate that here to bypass the “Comments to the Author” section, enter your conflict of interest statement in the “Confidential to Editor” section, and submit your "Accept" recommendation.

Reviewer #1: All comments have been addressed

Reviewer #2: All comments have been addressed

2. Is the manuscript technically sound, and do the data support the conclusions?

Reviewer #1: Yes

Reviewer #2: Yes

3. Has the statistical analysis been performed appropriately and rigorously? 

Reviewer #1: Yes

Reviewer #2: Yes

4. Have the authors made all data underlying the findings in their manuscript fully available?

Reviewer #1: Yes

Reviewer #2: Yes

5. Is the manuscript presented in an intelligible fashion and written in standard English?

Reviewer #1: Yes

Reviewer #2: No

6. Review Comments to the Author

Reviewer #1: Once again, excellent study. Good job with the improvements. Here are a few more suggestions.

Line 26 there is no need to repeat “(10%)”

114-115, note to authors and/editors to ensure that “-25” is on the same line

Line 178- “were weighed” is the correct grammatical form.

Line 332- “have”

Line 354- “which have”

Line 361- “which are”

Line 389- “decreases”

Line 401- rather than “activities,” would “properties” be a better word?

Line 412/413- “and to assess”

Reviewer #2: Manuscript should be prepared as a short paper and must check by native people.

Authors should update references in the introduction and discussion parts and explain the reasons of observed results.

7. PLOS authors have the option to publish the peer review history of their article (what does this mean?). If published, this will include your full peer review and any attached files.

Reviewer #1: **Yes: **Dr. Amina Moss

Reviewer #2: No

---

## [Author Response · Author response to Decision Letter 1]

18 Oct 2021

Dear Editor In-Chief,

Thank you very much for the reviewer’s comments to improve our manuscript (PONE-D-21-14702R1). According to the reviewer’s comments/suggestions including yours, all the required changes or revisions were carefully made in the 2nd revised manuscript. The corrections and answers are listed below:

[Reviewer #1]

Q1: Line 26. There is no need to repeat “(10%)”

A1: We corrected the sentence as the reviewer suggested (Line 26).

Q2. Line 114–115. Note to authors and/editors to ensure that “–25” is on the same line

A2: We dried the experimental diets after pelleting at “25℃” and stored the diets at” –25℃”. 

Q3. Line 178–. “were weight” is the correct grammatical form.

A3: We changed the sentence as the reviewer suggested (Line 178).

Q4. Line 332–. "have"

A4: We changed the sentence as the reviewer suggested (Line 331).

Q5. Line 354–. "which have"

A5: We changed the sentence (Line 351).

Q6. Line 361–. "which are"

A6: We changed the sentence as the reviewer suggested (Line 358).

Q7. Line 389–. "decreases"

A7: We changed the sentence as the reviewer suggested (Line 386).

Q8. Line 401–. Rather than "activities," would “properties” be a better word?

A8: We changed the sentence as the reviewer suggested (Line 399).

Q9. Line 412/413– “and to assess”

A9: We changed the sentence as the reviewer suggested (Line 409).

[Reviewer #2]

Q1. Manuscript should be prepared as a short paper and must check by native people.

A1. The manuscript was rechecked by an English native (essayreview.co.kr) one more time. We carefully revised the manuscript (PONE-D-21-14702R1) to improve readability. Please find the revised manuscript (R2).

Q2. Authors should update references in the introduction and discussion parts and explain the reasons of observed results.

A2. We agree with the reviewer’s comment. We updated references and carefully revised some sentences accordingly in introduction and discussion sections. Please check the revised manuscript.

---

## [Editor Report · Decision Letter 2]

8 Nov 2021

Digestibility of insect meals for Pacific white shrimp (Litopenaeus vannamei) and their performance for growth, feed utilization and immune responses

PONE-D-21-14702R2

Dear Dr. Lee,

We’re pleased to inform you that your manuscript has been judged scientifically suitable for publication and will be formally accepted for publication once it meets all outstanding technical requirements.

Kind regards,

Mahmoud A.O. Dawood, PhD

Academic Editor

PLOS ONE
---

## [Editor Report · Acceptance letter]

11 Nov 2021

PONE-D-21-14702R2 

Digestibility of insect meals for Pacific white shrimp (*Litopenaeus vannamei*) and their performance for growth, feed utilization and immune responses 

Dear Dr. Lee:

I'm pleased to inform you that your manuscript has been deemed suitable for publication in PLOS ONE. Congratulations! Your manuscript is now with our production department. 

Kind regards, 

on behalf of

Dr. Mahmoud A.O. Dawood 

Academic Editor

PLOS ONE